# OpenReview forum: "Microcanonical Hamiltonian Monte Carlo and the Helmholtz Theorem"
_TMLR — Under review for TMLR_

### Review · Reviewer_HzJ1 · 2026-05-22

**Summary Of Contributions:**

The paper studies Microcanonical Hamiltonian Monte Carlo (MCHMC) from a thermodynamic perspective. Its main contribution is to analyze whether MCHMC is microcanonical in the thermodynamic sense by showing that it satisfies the generalized Helmholtz theorem. The authors derive the relevant microcanonical partition functions, volume entropy, temperature, and conjugate thermodynamic variables for the MCHMC Hamiltonians, and they discuss the relation between microcanonical thermodynamic entropy and the information entropy of the posterior distribution. They further argue that random resampling of the momentum direction in MCHMC helps realize the monocyclicity requirement associated with the Helmholtz theorem, thereby giving a thermodynamic interpretation of this algorithmic component. These analytical claims are supported by numerical checks on a low-dimensional Gaussian target and on a cosmological inference problem using Type Ia supernova data.

A key strength of the paper is that it provides a coherent and technically motivated interpretation of an existing sampling method, connecting MCHMC, microcanonical ensembles, the Helmholtz theorem, and Bayesian inference within a single framework. I found the interpretation of momentum-direction resampling particularly useful, since it gives a thermodynamic rationale for an algorithmic component that is otherwise easy to view as a technical detail. The comparison between microcanonical and canonical MCMC methods is also valuable, especially the discussion of why canonical entropy has a more direct relation to the Shannon entropy of the posterior.

A limitation of the paper is that the contribution is primarily conceptual and foundational. The paper does not introduce a new sampling algorithm, and the empirical evaluation is mainly designed to validate the thermodynamic interpretation rather than to demonstrate practical gains in ML-oriented inference tasks. It is not yet fully clear whether the analysis leads to improved sampler design, diagnostics, tuning principles, or performance in practical machine-learning sampling problems.

**Additional Comments:**

I found the paper interesting and clearly written overall.  My main concern is not about the technical correctness of the core derivations, but about the positioning of the paper for TMLR. In its current form, the contribution is primarily foundational and thermodynamic, while the implications for machine-learning sampling practice remain somewhat indirect. I would encourage the authors to strengthen the discussion of what ML practitioners should take away from the analysis.

**Audience:**

Yes

**Audience Explanation:**

Some individuals in the TMLR audience would likely be interested in the paper, especially researchers working on Monte Carlo methods, Hamiltonian-based samplers, and physics-inspired machine learning. There are useful conceptual findings in the paper for readers interested in the foundations of sampling algorithms.

While some TMLR readers may find the findings interesting, the paper’s relevance to the broader TMLR audience is limited in its current form.

**Broader Impact Concerns:**

I do not identify any broader impact concerns for this work.

**Claims And Evidence:**

Yes

**Claims Explanation:**

The main claims of the submission are supported by accurate and reasonably convincing evidence. The central claim is that MCHMC satisfies the generalized Helmholtz theorem and can therefore be interpreted as microcanonical in a thermodynamic sense. This claim is supported first by analytical derivations where the authors derive the microcanonical partition functions, the corresponding volume entropy, temperature, and conjugate thermodynamic variables for the MCHMC Hamiltonians, and then verify the Helmholtz relation within this framework.

The analytical argument is further supported by numerical evidence. The authors test the relation on a low-dimensional Gaussian target, where the temperature estimated from MCHMC samples agrees with the theoretical and finite-difference calculations. They also repeat this check on a cosmological inference problem using Type Ia supernova data, where the sample-based temperature again agrees with the entropy-derivative estimate. These examples are not broad performance benchmarks, but they are appropriate for validating the specific theoretical claim made in the paper.

**Requested Changes:**

The paper is technically interesting and reasonably clear. However, Points 1–2 would significantly strengthen the paper's framing and I encourage the authors to address them and others as well.

1. The current contribution is mainly a thermodynamic interpretation of MCHMC. While this is conceptually interesting, it is not yet clear enough what the TMLR/ML audience learns from this analysis beyond the validation of MCHMC as a microcanonical method. The authors should explain more explicitly whether the Helmholtz-theorem perspective leads to practical or conceptual insight for ML sampling, for example in terms of sampler design, diagnostics, ergodicity, mixing, tuning, stability, or failure modes (any one of these). Without this connection, the paper’s relevance to TMLR remains somewhat indirect.

2. One of the most interesting points in the paper is the argument that random resampling of the momentum direction is related to the monocyclicity requirement associated with the Helmholtz theorem. However, the paper should make clearer what follows from this observation in practice. Does the thermodynamic argument imply anything about how often resampling should be performed, what can go wrong if resampling is insufficient, or whether this can be diagnosed through exploration, autocorrelation, ESS, or failure modes? Developing this point would make the theoretical contribution more useful for understanding MCHMC as a sampling algorithm.

3. The paper would benefit from a more explicit discussion of limitations, including dimensionality, entropy estimation and the low-dimensional nature of the numerical tests.

4. The paper should more clearly distinguish what is already known about MCHMC from what is newly contributed here, especially since the paper does not propose a new sampler but gives a thermodynamic/foundational interpretation of an existing one.

5. Please clarify the intended sign convention and make Eqs. (32), (34), and Table 1 consistent.

---

> ### Author Response · Authors · 2026-05-30
>
> Dear reviewer,
>
> Thank you for your review!
>
> In general, the aim of our paper is to shed a "thermodynamic light" on MCHMC. As
> always with interdisciplinary papers, it is a difficult question where to submit
> them. While our results offer very interesting insights (it does not get more
> fundamental than the first law of thermodynamics), the practical implications
> remain limited, as you say. We have nevertheless submitted to TMLR because we
> feel that the thermodynamic point of view on MCMC methods should be better known
> to practitioners from the stats and ML communities, and the presented results
> are an excellent example of this. We feel that this knowledge in itself is
> already of great value to those communities and that TMLR is thus the right
> address for our submission.
>
> At the same time, the two reviews have made us aware of the importance of the
> "random bounces", monocyclicity and ergodicity, which is why we have restructured
> the paper to include a subsection about this. With the connection to ergodicity,
> we have identified a more direct link to practical implications of the conditions
> for the Helmholtz theorem.
>
> Regarding your points, we have implemented the following changes:
> 1. & 2. See above and in sec. 3.1 of the revised manuscript. Additionally,
> we have reintroduced a new, volume-based microcanonical sampling algorithm that
> we had removed due to lack of space. It is a low-dimensional extension of MCHMC
> and as such relevant to practitioners, although for a low-dimensional inference
> problem one could probably simply use HMC.
> 3. We have added two paragraphs about limitations at the end of sections 3.2 &
> 3.4. Additionally, we have added an experiment that tests the measurements of
> the Gaussian in higher dimensions.
> 4. To better distinguish previous work and our contributions, we have
> rephrased the introduction as well as the conclusion.
> 5. Thank you for pointing this out. We've actually found a sign error in our
> calculations that we corrected, now they support our claims even better.
>
> If you have further points or if we have not sufficiently addressed your
> concerns, please let us know.
>
> Kind regards,
> the authors.

---

> > ### Comment · Reviewer_HzJ1 · 2026-06-03
> > **Response to authors’ revisions**
> >
> > Dear authors,
> >
> > Thank you for the detailed and thoughtful response, and for the substantial revisions to the manuscript.
> >
> > I appreciate that the authors have clarified the connection between random bounces, monocyclicity, and ergodicity, and that a dedicated discussion of these aspects has been added. This addresses my main concern about making the thermodynamic analysis more directly relevant to the behavior of the sampling algorithm. I also appreciate the added discussion of limitations. Overall, I find that the revised manuscript addresses my main concerns sufficiently.
> >
> > Based on the response and the described revisions, I am satisfied with the changes and support acceptance.

---

### Review · Reviewer_Zz48 · 2026-05-23

**Summary Of Contributions:**

This paper studies Microcanonical Hamiltonian Monte Carlo (MCHMC) from a thermodynamic/statistical-mechanical viewpoint. Its main contributions are, 1) deriving microcanonical partition functions for two Hamiltonians associated with MCHMC, 2) relating these constructions to Bayesian evidence and posterior marginalization, 3) analyzing the Helmholtz theorem for these systems, and 4) numerically checking the resulting temperature identities on a toy Gaussian example and a low-dimensional cosmological supernova inference problem.

**Audience:**

Yes

**Audience Explanation:**

The paper would likely interest a subset of TMLR readers working on Monte Carlo methods, Bayesian computation, probabilistic machine learning, and physics-inspired sampling algorithms. MCHMC and related microcanonical methods are active topics, and a careful thermodynamic interpretation could be valuable to researchers trying to understand when these algorithms are conceptually distinct from or equivalent to more standard HMC methods.

**Broader Impact Concerns:**

The paper is theoretical and concerns the interpretation of Monte Carlo algorithms, so its direct societal impact is likely limited.

**Claims And Evidence:**

No

**Claims Explanation:**

The paper’s argument shows that the idealized ensemble associated with the proposed Hamiltonian has the formal thermodynamic structure, not that every practical MCHMC algorithm automatically satisfies the theorem. The random bounce mechanism may help enforce exploration of the energy surface, but the paper does not prove irreducibility, ergodicity, or monocyclicity for general targets. Figure 1 is a useful illustration, but it does not justify the broader conclusion that the Helmholtz theorem is the theoretical reason for MCHMC’s superiority over earlier methods.

**Requested Changes:**

1. The paper should separate three claims:
a. The Hamiltonian defines a formal microcanonical ensemble
b. The ideal ensemble satisfies the Helmholtz theorem under suitable assumptions
c. A practical MCHMC chain samples this ensemble accurately.
The current text sometimes moves between these without enough justification.

2. Tone down or substantiate the claim that the Helmholtz theorem explains MCHMC’s superiority.

3. Include more details on sampler initialization, burn-in, number of resampling events, numerical integrator, step-size sensitivity, and convergence diagnostics.

4. In the explanation of Figure 2, the statement that the kinetic and potential energies plateau because the Euler-Lagrange equations minimize the action is not sufficiently justified and may be misleading.

---

> ### Author Response · Authors · 2026-05-30
>
> Dear reviewer,
>
> Thank you for your review!
>
> As far as we understand, your main points of criticism are a lack of clean
> structure and that the confidence of our claim about MCHMC's superiority
> compared to ESH is not proportional to the arguments that support it. We have
> addressed the latter by reformulating that the Helmholtz theorem merely offers
> an additional, fundamental perspective on this in a dedicated section (3.1)
> in the revised manuscript.
>
> In detail, we would like to address your points as follows:
> 1. Thank you for this suggestion. We have separated our contributions
> into different claims that are summarized in the newly formulated conclusion.
> Additionally, we have added transitions between these sections that
> should make these distinctions more clear.
> 2. See above.
> 3. We have added a new appendix that contains details about all our numerical
> experiments. Please note that this paper certainly does not aim to provide a
> comprehensive numerical test of the algorithm.
> 4. You're right, this is not generally true; it's easy to find counterexamples.
> We have added a corresponding disclaimer and reformulated, mentioning this as a
> possible explanation.
>
> If we have not sufficiently addressed something or if you would like to make
> additional points, please let us know. Thank you for taking the time to review
> our paper!
>
> Kind regards,
> the authors.

---

### Review · Reviewer_BXJ3 · 2026-06-23

**Summary Of Contributions:**

In this paper, the authors study the Microcanonical Hamiltonian Monte Carlo (MCHMC) algorithm proposed by Robnik et al. (2023) from a thermodynamic perspective. In particular, they demonstrate, both analytically and numerically, that the algorithm satisfies the Helmholtz theorem and additionally propose a volume-based microcanonical sampling algorithm.

Strengths:
- The paper provides a thermodynamic perspective on MCHMC, which may contribute to a deeper understanding of the algorithm and its properties.
- The mathematical derivations appear technically sound, as far as I can assess.

Weaknesses:
- The paper lacks sufficient motivation. It remains unclear why the presented analysis is important and what practical or theoretical benefits are gained from studying MCHMC from a thermodynamic perspective. Although an alternative sampling method is proposed, its advantages and potential applications are not adequately discussed. As a result, the paper's main contribution is hard to identify.

- The overall structure could be improved. Throughout the paper, it is often unclear why certain analyses are performed and what specific questions or hypotheses the authors aim to address. This issue is particularly evident in Section 4, which begins with "We will look at..." without clearly stating the purpose or the hypothesis being tested.

**Audience:**

Yes

**Audience Explanation:**

I believe this paper may be of interest to a (small) subset of the TMLR audience, particularly researchers with expertise in this exact area. However, its primary contribution is likely to provide additional insights for those already working on the topic, rather than offering broader benefits or appeal to the wider machine learning community.

**Claims And Evidence:**

Yes

**Claims Explanation:**

The authors claim that "the algorithm satisfies the Helmholtz theorem" which is shown analytically and numerical for multiple cases.

**Requested Changes:**

- Clearly state and emphasize the main contribution(s) of the paper, both in the introduction and throughout the manuscript.
- Strengthen the motivation by explaining why a thermodynamic perspective on MCHMC is useful, necessary, or expected to provide new insights.
- Expand the discussion in Section 2 regarding the relationship to Herzog et al. (2024), particularly clarifying similarities and differences between the macrocanonical and microcanonical viewpoints.
- In Equation (5), explicitly explain the meaning of the subscript and superscript (i).
- After Equation (6), replace "(see below)" with a more specific reference to the relevant section or equation.
- The sentence following Equation (6), "A different instance is the macrocanonical ensemble, ...", reads awkwardly and should be rephrased for clarity.
- After the statement following Equation (15), “Furthermore, it could inspire a new, volume-based, microcanonical sampling algorithm,” add a brief discussion explaining why such an algorithm would be interesting and what advantages it might offer.
- Begin Section 3 with a short paragrpah outlining the goals of the section and the steps of the analysis.
- Figure 2 needs to be better explained as I do not understand the purpose of the plots.
- In Section 4, clearly state the purpose of the investigation. Rather than beginning with "We will look at...", formulate a specific research question or hypothesis and explain how the analysis addresses it.

---

> ### Author Response · Authors · 2026-06-29
>
> Dear Reviewer,
>
> Thank you for your review!
>
> Our general motivation for this project is to study the thermodynamic traits of
> MCHMC. Initially, we did not have a specific goal in mind other than thoroughly
> understanding the algorithm from this perspective. In the past, this point of
> view has inspired new convergence criteria for MCMC samplers (Röver, 2023b) and
> the macrocanonical Avalanche sampler (Herzog, 2024).  As we say in our reply
> to reviewer HzJ1, we believe that this study in itself and the fundamental
> knowledge gained from it are already of great value to the community, even if
> the practical implications are limited.
>
> To our minds, the most interesting findings following from this open-ended
> analysis are the following:
> - the fact that MCHMC fulfils the first law of thermodynamics in the form of the
>   Helmholtz theorem,
> - the conceptual connection between the Helmholtz theorem and ergodicity which
>   is related to how MCHMC improves over its predecessor,
> - the new sampling algorithm VMCHMC (although its usefulness in practice may be
>   limited),
> - the demonstration that no microcanonical sampling algorithm will have an
>   associated thermodynamic entropy that is consistent with Shannon's entropy
>   of the posterior.
> We have modified the manuscript to emphasize these more clearly. So far, VMCHMC
> has remained in the appendix since we did not want to exceed 12 pages of main
> text. If you deem it important enough to be moved to the main body, we would be
> happy to do so.
>
> Thank you for your comments on how the structure of the manuscript may be
> improved. We have implemented them, see the newly revised version. In
> particular, we have added a short introduction to Section 3 (revised version)
> explaining our approach to testing the Helmholtz theorem. Additionally, we have
> separated the part about the thermodynamic nature of MCMC methods into its own
> section since it its contents are distinct from the other discussion. We have
> furthermore explained figure 2 in more detail.
>
> Please let us know if we have not sufficiently addressed some of your points or
> if you would like to raise additional concerns.
>
> Kind regards.
> The authors.
>
>
> references as in the paper